# Enhanced IR Radiative Cooling of Silver Coated PA Textile

**DOI:** 10.3390/polym14010147

**Published:** 2021-12-31

**Authors:** Xiaoyu Xie, Yang Liu, Ying Zhu, Zhao Xu, Yanping Liu, Dengteng Ge, Lili Yang

**Affiliations:** 1State Key Laboratory for Modification of Chemical Fibers and Polymer Materials, College of Materials Science and Engineering, Donghua University, Shanghai 201620, China; 2190414@mail.dhu.edu.cn; 2State Key Laboratory for Modification of Chemical Fibers and Polymer Materials, Institute of Functional Materials, Donghua University, Shanghai 201620, China; yangliu@mail.edu.dhu.cn (Y.L.); xuzhao@mail.dhu.edu.cn (Z.X.); dengteng@dhu.edu.cn (D.G.); 3College of Textiles, Donghua University, Shanghai 201620, China; 2200272@mail.dhu.edu.cn (Y.Z.); liuyp@dhu.edu.cn (Y.L.)

**Keywords:** smart textile, silver coating, radiative cooling, multi-order reflection

## Abstract

Smart textile with IR radiative cooling is of paramount importance for reducing energy consumption and improving the thermal comfort of individuals. However, wearable textile via facile methods for indoor/outdoor thermal management is still challenging. Here we present a novel simple, yet effective method for versatile thermal management via silver-coated polyamide (PA) textile. Infrared transmittance of coated fabric is greatly enhanced by 150% due to the multi-order reflection of silver coating. Based on their IR radiative cooling, indoors and outdoors, the skin surface temperature is lower by 1.1 and 0.9 °C than normal PA cloth, allowing the textile to be used in multiple environments. Moreover, the coated fabric is capable of active warming up under low voltage, which can be used in low-temperature conditions. These promising results exemplify the practicability of using silver-coated textile as a personal thermal management cloth in versatile environments.

## 1. Introduction

When the body’s energy metabolism and heat dissipation to the surrounding environment reach a balance, it will feel comfortable [1]. However, in extreme environmental conditions, if the heat generated by the human body is out of balance with the environment, it will feel uncomfortable, affect people’s physical and mental health [2], and even lead to serious physical therapy accidents. Space energy and cooling systems can effectively alleviate this problem, but at the same time they will consume excessive energy [3,4,5] and cause global warming and extreme weather [6,7]. “Personal thermal management” has become an effective solution to the current problems [8,9]. Its main idea is to realize the technology of heating or cooling the human body’s local environment, which can avoid wasting excess electricity on heating or cooling the entire building, reduce energy consumption, and improve energy efficiency while improving the thermal comfort of the human body. When a person is in an indoor environment, the body’s surface temperature is maintained at 34 °C, the radiated infrared (IR) wavelength range is 7–14 μm, while traditional textiles have the characteristics of high infrared absorption and low infrared emission [10], which seriously inhibits heat loss in hot summer. Therefore, a kind of “radiation cooling” fabric is needed to improve the comfort of the human body in a hot environment.

Existing radiant cooling fabrics increase heat loss mainly by improving the thermal radiation transmittance or solar reflectance of the fabric. The former uses infrared transparent and visible opaque polymers to provide radiant cooling through thermal radiation emitted by the human body to the environment. Nanoporous polyethyelene (nanoPE) is transparent to mid-infrared radiation but opaque to visible light; however, it shows poor air permeability. Microscaled holes were created using microneeedle punching and then hydrophilic agent polydopamine was used to improve the air permeability [11]. Besides, ZnO nanoparticles were embedded in the nanoPE film to improve the solar reflectivity via melt-pressing [12]. Similarly, SiO_2_ spheres were comprised into polyamide 6 nanofiber membrane via electrospinning and exhibited enhanced IR radiation [13]. All these achieved a satisfying cooling capability indoors or outdoors. Nevertheless, these materials appear as membrane and are not suitable for general textile. The second type of solution to increase the solar reflectance of fabrics mainly includes metal laminate or deposition [14,15,16], and fiber porosity improvement [17]. Obviously, this type of material has shortcomings such as poor air permeability and complex processing technology, which makes it difficult to promote and apply. Therefore, a novel strategy to improve radiant cooling for general textile via facile method remains a challenge.

As illustrated in Figure 1, the human body emits mid-infrared radiation; however, conventional fabric (such as PA) shows low infrared transmittance, which suppresses the IR radiative heat dissipation. Here, we present a high infrared transmittance fabric for personal thermal management based on multistage infrared reflection of silver-coated fiber. Highly efficient heat dissipation is achieved while the original wearing comfort is maintained. We believe that this Ag@PA textile provides a new idea for smart textile.

## 2. Materials and Methods

### 2.1. Materials and Preparation

PA fiber bundles (Nylon-66, 280 D) were purchased from Qingdao Tianyin Textile Technology Co., Ltd., Qingdao, China. Ag@PA fiber bundles (280 D) were prepared from the electroless plating on PA fiber bundles using a solution of 7 g/L silver nitrate, 3 g/L ammonia, 15 g/L glucose, and 2 g/L catalyst hexamethylenetetramine for 40 min. Silver nanowires dispersion (AgNWs with a diameter of 30 nm and a length of 30 μm, anhydrous ethanol with a concentration of 3 mg/mL) was purchased from Shanghai Bohan Chemical Technology Co., LTD., Shanghai, China. PA fabric (58 g/m^2^) and Ag@PA fabric (66 g/m^2^) was directly woven from PA bundles and Ag@PA fiber bundles on Stoll ADF 530W. AgNW-PA cloth (60 g/m^2^) was achieved by dip-coating PA cloth in Ag NWs solution and vacuum drying for 6 h. Infrared transparent and visible light opaque (nanoPE) film (thickness of 16 μm) was purchased from Asahi Kasei Company, Tokyo, Japan.

### 2.2. Characterization

Scanning electron microscope (SEM) images and EDS mapping were taken by TESCAN (MAIA3, Brno, Czech Republic). The XRD patterns were characterized by X-ray diffractometer (Bruker D8, Billerica, Germany). IR transmittance (τ) was measured by FT-IR spectrometer (Thermo Scientific Nicolet iS 50, Waltham, MA, USA) using an integrating sphere accesssory (PIKE, Madison, WI, USA). The IR photo was taken by an infrared thermal imager (Flir T62101, Hamburg, Germany). The infrared transmission energy was measured by an infrared thermal imager and ResearchIR 4 software (research ir max 4.11.35, Flir, Hamburg, Germany). A fiber optic spectrometer (USB2000 spectrometer, Ocean Optics, Dunedin, FL, USA) was used to measure the visible light reflectance and transmittance of fabrics. The temperature was measured by the contact thermocouple with digital thermometer (GM1361, Maoyuan Technology Company, Shenzhen, China).

IR reflection of fiber bundle: the fiber bundles were tightly twined side by side on a white paper-made cardboard. The temperature of surface heat source (10 × 20 cm^2^) was set to 70 °C. The distance between bottom of the substrate and the heat source was maintained at 2 cm. The effective reflection area was changed via the rotation angle α between the substrate and the heat source. An infrared thermal imager was placed horizontally to measure the IR reflected energy. IR radiation of fiber bundle: the fiber bundles were arranged closely on the PE substrate in a single and three layers. The surface heat source was set at 70 °C, and the relative height of the substrate and heat source was maintained at 6 cm. The IR thermal imager was perpendicular to the horizontal plane of the substrate. IR emission of fiber bundle: the fiber bundles were tightly twined side by side on white cardboard, and the surface heat source temperature was set to 50 °C. The cardboard was attached to the heat source and the temperature kept constant after 5 min. The infrared thermal imager was set to the temperature measurement mode, and three different measurement points were taken to measure the temperature for three times. The ambient temperature and the actual fiber bundle temperature were generated by the heat of the contact thermocouple. The emissivity of the two fiber bundles can be calculated by:(1)$$\varepsilon =\frac{{\left(T_{0}^{\prime }\right)}^{n}-T_{u}^{n}}{T_{0}^{n}-T_{u}^{n}}\;$$
where $T_{0}^{\prime }$ is the temperature displayed by the infrared camera, *T_u_* is the ambient temperature, *T*_0_ is the actual temperature of the fiber bundle, and the value of *n* is related to the working band of the thermal imager, for the band range of 8–13 μm, the value of *n* is 4.09.

Joule Heating tests: The cloth sample was cut into 3 cm × 3 cm, and copper tape was pasted on both ends of the sample for electrical contact. The voltage was provided by Mestek DP152 DC stabilized power supply. Temperatures were monitored by a thermocouple (diameter 0.3 mm, type K) connected to a temperature controller.

IR transmission simulation: COMSOL Multiphysics software was used to simulate the infrared energy transmission. A two-dimensional model was chosen, and ray optics → geometric optics (GOP) for the physical field was selected. The geometry model was circular and formed into an array (double layer). The materials were set to Ag and PA, respectively. The wavelength of ray was set to 9.5 μm.

## 3. Results

### 3.1. Morphologies of Silver-Coated Fiber

As a typical commercial fiber, PA fiber is here presented in Figure 1A(i) with a diameter of ~20 μm. After the electroless plating, fibers remain their original morphologies, i.e., Ag@PA fiber (Figure 1A(ii)). The diameter of Ag@PA fiber is almost unchanged, indicating the uniform and very thin silver layer. The EDS element distribution images of Ag@PA fiber (Figure 1B) present the full coverage of Ag on the PA fiber surface. The high-resolution cross-sectional SEM image of Ag@PA fiber shown in Appendix A and large-scaled EDS image of Ag@PA fabric in Appendix A confirm the uniform distribution of Ag on the surface. The XRD patterns of PA fibers and Ag@PA fibers are also shown in Appendix A. The diffraction peaks at 20 and 24° or merged 22.4° correspond to the characteristic peaks of PA fiber. The XRD peaks appearing at 38.22, 43.51, 65.34, and 76.40° indicate the (111), (200), (220), and (311) planes of face centered cubic structure of silver (JCPDS No. 87–0719). The XRD results also demonstrate that silver is successfully coated on the surface of PA fiber.

### 3.2. IR Properties of Ag@PA Fiber Bundle

In order to study the infrared properties of Ag@PA fabric, the infrared properties of Ag@PA fiber bundle are firstly investigated, including IR reflection, transmission, and emission. Figure 2A presents a schematic diagram for the infrared reflection tests. The fiber bundles are tightly twined side by side on the white cardboard (as a low IR reflective material). Figure 2B shows the IR images of two bundles at rotation angle α between the cardboard and heat source of 90 and 30°. The Ag@PA fiber bundle seems brighter than PA fiber. When the rotation angle α gradually decreases from 90°, the reflection energy increases (as shown in Figure 2C). The reflection energy of Ag@PA bundle remains higher than that of the PA bundle, indicating the high reflection of silver coating. Figure 2D shows a schematic diagram for testing infrared transmission performance. The fiber bundles are twined on the PE substrate with single and three layers, respectively. Figure 2E,F exhibit the infrared radiation curve in the vertical direction of fiber arrangement. It can be seen that the PA fiber bundle maintains low IR radiation in both the monolayer and three-layer condition. In contrast, the IR radiation of Ag@PA fiber bundle demonstrates high–low values alternately. Comparing with the bundle image, it is obvious that high IR radiation comes from the edge of the Ag@PA fiber bundle, which is probably due to the high IR reflectivity of the silver coating. Figure 2G shows the IR transmittance of the two fiber bundles using an integrating sphere accessory. The IR transmittance comparison confirms the huge difference before and after the silver coating. Using the experimental setup shown in Appendix A and based on the Equation (1), the emissivity ε of Ag@PA fiber bundle and PA fiber bundle is calculated to be 0.69 and 0.87, respectively, indicating the lower IR absorption of Ag@PA fiber bundle.

In order to further describe the IR penetration path of fiber bundles and understand the difference of IR transmittance between different fiber bundles, the infrared transmittance process is simulated via COMSOL Multiphysics. Figure 2H and Figure 2I show the paths of IR lights after the incidence on Ag@PA fiber bundle and PA bundle, respectively. The incident infrared light is absorbed greatly in the PA fiber bundle, and almost no transmission is observed. Ag@PA fiber bundles greatly increase the transmitted infrared beams due to the multi-level reflection of infrared rays on the multilayer silver-plated fiber.

### 3.3. Infrared Properties of Ag@PA Fabrics

PA fabric and Ag@PA fabric are woven using bundles under the same condition. Figure 3A,C demonstrate the same weave structure of two fabrics. IR thermal images of one-layered or double-layered fabric are taken in order to compare the IT transmission visually. Figure 3D shows that PA fabric has high infrared transmission energy only at the cavity of fiber bundle edge. On the contrary, the infrared transmission energy of Ag@PA fabric (Figure 3B) is greatly enhanced whether it is in a single-layer or double-layer state. Figure 3E summarizes the IR energy flux at different temperatures of heat source (40, 60, and 80 °C). It is indicated that the IR transmission radiance of Ag@PA fabric is greater than that of PA fabric by 20–50%. The IR transmission using a Fourier Transform infrared (FTIR) spectrometer with a diffuse gold integrating sphere and IR radiance at different heat sources are also studied. Figure 3F shows that the infrared transmittance of Ag@PA fabric in the 2.5–12 μm infrared band is around 40%, about 2.5 times of that of PA fabric.

### 3.4. IR Adaptive Applications of Ag@PA Textile

Personal thermal management includes the applications of IR-adaptive textile in indoor and outdoor environments. In order to compare the IR-adaptive properties, PA cloth doped with silver nanowires (AgNW-PA cloth) was also prepared and tested. As shown in Figure 4A, three different fabrics were wrapped on the forearm in double layers, which were then exposed in the lab. The thermal images were also taken and all samples were in thermal equilibrium before imaging. Moreover, the infrared radiation energy penetrating the fabric was also tested. Compared with three fabrics, the skin surface temperature beneath the PA cloth was the highest, and the temperature difference between two sides of PA cloth was the greatest, suggesting the excellent thermal insulation of PA cloth. In contrast, the thermal images show that the area covered by Ag@PA cloth is in a “hot “ state, indicating that the infrared transmission energy value in this area is large, which is beneficial to the heat dissipation of the human body. In the indoor environment, the temperature of the Ag@PA cloth covered skin only increased by 0.4 °C compared with the bare skin (34.1 °C), which achieved a 1.1 °C drop (35.6−34.5 °C) in temperature compared to the traditional PA cloth.

People inevitably need to be exposed to high-temperature outdoor environments during daily life, while an outer high temperature will cause the Ag@PA cloth to heat up more quickly. In order to decrease the sunlight radiation in visible band, as shown in Figure 4C,E, an infrared transparent and visible light opaque (nanoPE) film was knitted on the surface of Ag@PA cloth, i.e., PE/Ag@PA cloth. This new fabric can not only maintain the high infrared transmittance of the original fabric, but also reflect visible light efficiently (as shown in Appendix A). The IR thermal images of different fabrics under sunlight were taken and the surface temperatures were tested. From the results in Figure 4D, the skin temperature beneath the PE/Ag@PA cloth was lower by 0.9 °C than that of traditional PA cloth with the outdoor sunlight radiation. The skin temperature beneath the Ag@PA cloth was close to that of PA cloth due to the great visible radiation absorption. These results prove that our fabric can be used both in indoor and outdoor environments.

This Ag@PA cloth is not only capable of passive heat dissipation but also active warming up in cold environments. One piece of Ag@PA cloth with area of 3 cm × 3 cm was clamped and copper electrodes were pasted on both ends. The temperature was measured by a thermal couple in close contact with the sample. As shown in Figure 4F, when only 1.5 V was applied, the temperature of the fabric can rise frequently. Figure 4G indicates the temperature change versus time when the cloth was applied with different voltages. The function of Joule heating in cold weather further complements the high radiation cooling described previously.

## 4. Conclusions

In summary, we have demonstrated a design for versatile personal thermal management fabric based on IR radiation cooling and electrical Joule heating. A uniform silver layer was coated on PA fiber via an electroless plating method. The IR properties of coated bundles through experiments and simulations confirmed that the Ag@PA fiber bundle has higher infrared reflectance and lower infrared absorption rate than the PA fiber bundle. The knitted fabric from Ag@PA fiber bundle showed enhanced IR transmittance by 150% due to the multi-order reflection of silver coating on the surface. Compared with traditional PA textiles, the skin temperature can be lowered by up to 1.1 °C indoor and 0.9 °C outdoor, realizing the “passive cooling” of the human body. In the cold winter, Joule heating can also be used to obtain additional heat to achieve “active heat preservation”. In addition, compared with ordinary textiles, Ag@PA cloth retains the advantages of light weight, breathability, and durability. We believe that this excellent high-infrared permeable textile can reduce people’s demand for indoor refrigeration, reduce fossil energy consumption, and provide an idea for alleviating global climate problems.

## Data Availability

The raw data presented in this study are available on request from the corresponding author.

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
