# Peer review of "Enhanced IR Radiative Cooling of Silver Coated PA Textile"

_polymers, 2021, doi:10.3390/polym14010147_

Round 1

Reviewer 1 Report

The article entitled “Enhanced IR radiative cooling of silver coated PA textile” study an interesting subject as the PA textiles with IR radiative cooling. After  revision the manuscript can be published. 

  1. In figures 2 and 3 are shown the infrared spectra. I think the authors have to introduce in the text the bands that appear as position and interpretation. Also, a correlation with properties is expected.
  2. Figure 4 A Optical and IR thermal images of three kinds of fabrics wrapped around forearm in 204 indoor and Figure 4C Optical and IR thermal 205 images of three kinds of fabrics wrapped around forearm outdoor present indoor and outdoor measurements. The authors must explain why there are so big differences between the two measurements. Theoretically speaking a property like the one studied should be preserved in both cases. The second question is why in Figure 4C are differences between the upper and lower parts of the material. How uniform is the filing of the film on the PA?
  1. Fig. S3 Visible transmittance and reflection of different fabrics must be added in the manuscript.
  2. The manuscript and references must be revised in accordance with the rules of the journal.

Author Response

Reviewer 1: In figures 2 and 3 are shown the infrared spectra. I think the authors have to introduce in the text the bands that appear as position and interpretation. Also, a correlation with properties is expected.

Response: Yes, IR transmittance of two kinds of bundles or fabrics were characterized using the integrating sphere accessory in Figure 2G and Figure 3F. Here, the IR tests are performed to compare the IR transmittance since the radiated IR wavelength range of human body is 7~14 μm. With the same thickness and area during testing, the IR transmittance of Ag@PA is distinctly higher than PA.

2.Figure 4 A Optical and IR thermal images of three kinds of fabrics wrapped around forearm in 204 indoor and Figure 4C Optical and IR thermal 205 images of three kinds of fabrics wrapped around forearm outdoor present indoor and outdoor measurements. The authors must explain why there are so big differences between the two measurements. Theoretically speaking a property like the one studied should be preserved in both cases.

Response: The differences of IR thermal images in indoor and outdoor environments mainly originate from the different hot sources in two conditions. When human body is in the room (indoor), the temperature of body is higher than that of room environment. Thus the image recorded by IR camera mainly comes from the IR radiated by human body. Accordingly, under the IR camera, the Ag@PA fabric shows higher energy due to its higher IR transmittance, as shown in Figure 4A. In contrast, when human body is out of the lab and under strong sunlights (outdoor). The radiation from sunlights is greatly higher than that of human body. Thus IR radiation energy from the outdoor environment plays the dominant role. Accordingly, the IR image recorded by IR camera mainly comes from the reflected IR from environment. Moreover, there are two distinct parts (upper part and lower part) in the IR images of Ag@PA fabric and PE/Ag@PA fabric. The reason for this will be available in the next question.

The second question is why in Figure 4C are differences between the upper and lower parts of the material.

Response: This is a good question which is in agreement with the last question. In the outdoor in summer, the ground which absorbed great sun lights gives strong infrared radiation to the surface of fabrics. For the Ag@PA fabric and PE/Ag@PA fabric, IR radiation is reflected by the silver coating surface and finally captured by the IR camera. Therefore, in the thermal image (Figure 4C), the lower part of the Ag@PA fabric will appear as an obvious "high energy" area, and the upper half will appear as a "low energy" area. In contrast, PA fabrics have high IR absorbance and can hardly reflect ground radiation. Therefore, the overall color of the IR image of PA fabric is the same.

According to the above analysis, the temperature from the IR camera is affected by the environments. Thus the temperature is measured by the contact thermocouple with digital thermometer (GM1361, Shenzhen Maoyuan Technology Company). 

How uniform is the filing of the film on the PA?

Response: Ag@PA fiber bundles are prepared from the electroless plating process on PA bundles using a mixed solution. According to the previous reports about electroless plating of metals on fibers (J. Appl. Polymer Sci. 2012, 124: 1912-1918; Materials, 2018, 11: 2033 et al.), the coating can be very uniform due to the control of treatment time, temperature, concentration et al.

In our manuscript, Figure 1 shows the SEM and EDS images of single fiber. In order to evaluate the distribution of silver coating on the fiber, we also added the cross-sectional SEM images as shown in Fig. S1 in the supporting information. The silver coating seems uniform on the single fiber.

Fig. S1 Cross-sectional SEM images of Ag@PA fiber.

Furthermore, in order to evaluate the distribution of silver coating in large scale, the SEM and EDS images of large scaled Ag@PA fabric are added in the supporting information as Fig. S2. From the EDS image of Ag element, the silver coating in a very large scale is uniform.

Fig. S2 SEM and EDS images of Ag@PA fabric. Scale bar: 500 μm.

Fig. S3 Visible transmittance and reflection of different fabrics must be added in the manuscript.

Response: The visible transmittance and reflection of different fabrics are shown in the supporting information as Fig. S5 on page 3.

The manuscript and references must be revised in accordance with the rules of the journal.

Response: We have revised the format of manuscript and references according to the rules of this journal.

Reviewer 2 Report

Dear Authors!

Сomments:

The introduction does not provide information on the current state of the available approaches to solving the problem. More than half of the literature refers to general issues that are not related to the problem posed by the authors. The solution to the problem is partially reflected in publications 11-15, but they are simply listed, and the essence of these approaches is not disclosed.

Why is silver chosen as a coating agent? How economically justified is it in relation to the obtained effects?

It is better to take an SEM image of the fiber cross-section in order to evaluate the thickness of the resulting silver layer, as well as the EDS of the longitudinal section of the fiber, in order to see the character of the distribution of silver along the fiber length. SEM of the longitudinal section at high magnifications would also help to judge the character of the distribution of silver particles on the fiber surface.

The use of XRD to provide the availability of silver is unjustified, since it does not provide any additional information.

Author Response

Reviewer 2:

The introduction does not provide information on the current state of the available approaches to solving the problem. More than half of the literature refers to general issues that are not related to the problem posed by the authors. The solution to the problem is partially reflected in publications 11-15, but they are simply listed, and the essence of these approaches is not disclosed.

Response: Thank you for your reminding. We have deleted some description about the general issues and added some detailed expression about the reported strategies in the Introduction part as follows:

“The former uses infrared transparent and visible opaque polymers to provide radiant cooling through thermal radiation emitted by the human body to the environment. Nanoporous polyethyelene (nanoPE) is transparent to mid-infrared radiation but opaque to visible light, however it shows poor air permeability. Microscaled holes were created using microneeedle punching and then hydrophilic agent polydopamine was used to improve the air permeability [11]. Besides, ZnO nanoparticles were embedded in the nanoPE film to improve the solar reflectivity via melt-pressing [12]. Similarly, SiO2 spheres were comprised into polyamide 6 nanofiber membrane via electrospinning and exhibited enhanced IR radiation [13]. All these achieved a satisfying cooling capability indoors or outdoors. Nevertheless, these materials appear as membrane, not suitable for general textile. The second type of solutions to increase the solar reflectance of fabrics mainly includes: metal laminate or deposition [14-16], and fiber porosity improvement [17]. Obviously, this type of material has shortcomings such as poor air permeability, and complex processing technology, which makes it difficult to promote and apply. Therefore, novel strategy to improve radiant cooling for general textile via facile method is still challenging.”

Why is silver chosen as a coating agent? How economically justified is it in relation to the obtained effects?

Response: In previous studies, silver has been coated on the fiber surface due to its excellent antibacterial property (Adv. Mater. Research 2011, 175: 687-690) or electromagnetic shielding (Appl. Mech. Mater. 2013, 239: 310-313). Furthermore, the cost of silver is moderate while the cheaper metals (such as Al, Ni) have lower corrosion resistance or oxidation resistance. Thirdly, in our research, silver coating shows high IR reflectivity and low IR absorbance.

It is better to take an SEM image of the fiber cross-section in order to evaluate the thickness of the resulting silver layer, as well as the EDS of the longitudinal section of the fiber, in order to see the character of the distribution of silver along the fiber length. SEM of the longitudinal section at high magnifications would also help to judge the character of the distribution of silver particles on the fiber surface.

Response: Yes, we added the cross-sectional SEM images to evaluate the thickness of silver coating. As shown in Fig. S1 in the supporting information, the thickness is about 1-2 μm. Moreover, the silver coating is uniform on the surface of fiber. In order to tell the distribution of silver coating in a large scale, we also added the EDS images of Ag@PA fabric as shown in Fig. S2

Fig. S1 Cross-sectional SEM images of Ag@PA fiber.

Fig. S2 SEM and EDS images of Ag@PA fabric. Scale bar: 500 μm.

The use of XRD to provide the availability of silver is unjustified, since it does not provide any additional information.

Response: The XRD spectra shown in Fig. S3 confirms the crystalline structure of silver coating on the surface of PA fiber.

Round 2

Reviewer 2 Report

The paper could be accepted for publication.